# Antibacterial Activity of Four Plant Extracts Extracted from Traditional Chinese Medicinal Plants against *Listeria monocytogenes, Escherichia coli,* and *Salmonella enterica* subsp. *enterica* serovar Enteritidis

**DOI:** 10.3390/microorganisms8060962

**Published:** 2020-06-26

**Authors:** R.L. McMurray, M.E.E. Ball, M.M. Tunney, N. Corcionivoschi, C. Situ

**Affiliations:** 1School of Biological Sciences, Queen’s University Belfast, Belfast BT9 5DL, Northern Ireland, UK; 2Agri-Food and Biosciences Institute, Hillsborough BT26 6DR, Northern Ireland, UK; elizabeth.ball@afbini.gov.uk; 3School of Pharmacy, Queen’s University Belfast, Belfast BT9 7BL, Northern Ireland, UK; m.tunney@qub.ac.uk; 4Agri-Food and Biosciences Institute, Veterinary Sciences Division, Belfast BT4 3SD, Northern Ireland, UK; nicolae.corcionivoschi@afbini.gov.uk; 5Institute for Global Food Security, Queen’s University Belfast, Belfast BT9 5DL, Northern Ireland, UK; c.situ@qub.ac.uk

**Keywords:** antibiotic resistance, pathogens, plant extracts, in vitro model, time–kill assay, broth microdilution, antimicrobial susceptibility, broiler, digest, chicken

## Abstract

The worldwide ethnobotanical use of four investigated plants indicates antibacterial properties. The aim of this study was to screen and determine significant antibacterial activity of four plant extracts in vitro and in a poultry digest model. Using broth microdilution, the concentrations at which four plant extracts inhibited *Listeria monocytogenes, Salmonella enteritidis,* and *Escherichia coli* over 24 h was determined. *Agrimonia pilosa* Ledeb*, Iris domestica* (L.) Goldblatt and Mabb*, Anemone chinensis* Bunge, and *Smilax glabra* Roxb all exhibited a minimum inhibitory concentration (MIC) of 62.5 mg/L and a minimum bactericidal concentration (MBC) of 500 mg/L against one pathogen. *A. pilosa* Ledeb was the most effective against *L. monocytogenes* and *E. coli* with the exception of *S. enteritidis*, for which *A. chinensis* Bunge was the most effective. Time–kills of *A. pilosa* Ledeb and *A. chinensis* Bunge against *L. monocytogenes, E. coli* and *S. enteritidis* incubated in poultry cecum were used to determine bactericidal activity of these plant extracts. *A. chinensis* Bunge, significantly reduced *S. enteritidis* by ≥ 99.99% within 6 h. *A. pilosa Ledeb* exhibited effective significant bactericidal activity within 4 h against *L. monocytogenes* and *E. coli*. This paper highlights the potential of these plant extracts to control pathogens commonly found in the poultry gastrointestinal tract.

## 1. Introduction

Antibiotic resistance is currently a global concern and significant research is taking place to tackle this issue [1,2], including research into the use of antibiotics in animal production [3]. Poultry farming is a rapidly growing global industry^3^. In many countries outside of Europe antibiotics are used as poultry feed additives at subtherapeutic levels over prolonged periods to promote growth and control gastrointestinal infections in the flock [4,5]. While there are legitimate therapeutic reasons for antibiotic use in poultry farming there are concerns about the overuse and misuse of antibiotics [3]. A link between subtherapeutic application of broad-spectrum tetracyclines in agriculture and the development of acquired-resistant human isolates led to a ban on the use of tetracyclines for growth promotion in Europe in the early 1970s [6]. Acquired resistance is the major contributor to baseline resistance and is a major threat to the continued success of antibiotics [7]. Due to the alarming rate of increased acquired antibiotic resistance in pathogens it is imperative that researchers explore other safe and sustainable alternatives to antibiotics. Alternatives to antibiotics could be used in livestock production to maintain production performance and control infections caused by pathogens commonly found in the poultry gastrointestinal tract. For example, *Escherichia, Salmonella,* and *Listeria* [8,9,10].

Numerous medicinal plants have been utilized as traditional medicines globally [11] for human therapeutic use to treat diseases of pathogenic origin [12]. Plant extracts consist of numerous bioactive compounds including polyphenols, terpenes, and phytosterols [13] and exhibit multiple modes of action to inhibit or kill bacteria [14]. An accumulating quantity of research has demonstrated that many plants used in traditional Chinese medicine have antibacterial activity both in vitro and in vivo in poultry production [15]. More specifically, the antibacterial properties of solvent extracts of *Agrimonia pilosa* Ledeb, *Smilax glabra* Roxb, *Anemone chinensis* Bunge, and *Iris domestica* (L.) Goldblatt and Mabb have been documented through in vitro screening [16,17,18,19]. Aqueous extraction methods are inexpensive and the bioactive compounds are less toxic and therefore more suitable for use as a poultry feed supplement. Furthermore, the antibacterial activity of the aqueous extract of *Anemone chinensis* Bunge exhibited antibacterial activity against *Staphylococcus aureus* during disc diffusion, highlighting its antibacterial properties [20]. *A. pilosa* Ledeb, *S. glabra* Roxb, *A. chinensis* Bunge, and *I. domestica* (L.) Goldblatt and Mabb are listed in the Chinese Pharmacopoeia [21] and are used to treat infections in many parts of the world. However, to our knowledge, no report exists on the bactericidal activity of these aqueous plant extracts in an in vitro poultry digest model. It is necessary to establish scientific evidence for the bactericidal activity of plant extracts as they may provide a source for the development of novel antibiotics.

*Agrimonia pilosa* Ledeb, *S. glabra* Roxb, *A. chinensis* Bunge, and *I. domestica* (L.) Goldblatt and Mabb are hardy, perennial plants and can therefore be grown in abundance in Europe, the UK, and Asia [22,23,24,25] with minimum maintenance. This makes them suitable as accessible sources of novel antibiotics. These plants are native to different regions. For example, *A. pilosa* Ledeb is native to Eastern Europe, China, Korea, and Japan. *A. chinensis* Bunge is native to East Russia, China, and Korea. *I. domestica* (L.) Goldblatt and Mabb is native across regions from Himalaya to Japan and Philippines and has been successfully introduced to the USA. *S. glabra* Roxb is native to Southeast Asia.

The antibacterial properties of plant extracts of *A. pilosa* Ledeb, *S. glabra* Roxb, *A. chinensis* Bunge, and *I. domestica* (L.) Goldblatt and Mabb, commonly used in traditional Chinese medicine to treat diseases of pathogenic origin are presented in this paper against three pathogens commonly found in the poultry gastrointestinal tract: *Escherichia coli, Salmonella enteritidis,* and *Listeria monocytogenes*. The aim of this research was to determine the antibacterial properties of these plant extracts and to evaluate the most effective bactericidal effect using an in vitro poultry digest model.

## 2. Materials and Methods

### 2.1. Preparation of Reference Strains and Clinical Isolates

Reference strains and clinical isolates were obtained as frozen stocks from Agri-Food and Biosciences Institute (AFBI), the National Collection of Type Cultures (NCTC), the American Type Culture Collection (ATCC), and Queen’s University Belfast (QUB) from several sources (Table 1). Each bacterium was identified using 16S PCR, Sanger sequencing, and BLAST analysis as per manufacturer’s instructions for using MyTaq™ Red Mix [26]. Gram staining was also used to confirm the identity of bacteria [27]. *L. monocytogenes, Salmonella enterica* subsp. *enterica* serovar Enteritidis*,* and *E. coli* were selected for antimicrobial susceptibility testing because they are pathogens that cause foodborne diseases and are commonly found in the intestinal tract of poultry [8,9,10].

### 2.2. Preparation of Plant Extracts

Four dry plant samples were purchased from Tong Ren Tang (Beijing). These included: the herb of *Agrimonia pilosa* Ledeb, the tuber of *Smilax glabra* Roxb, the rhizome of *Iris domestica* (L.) Goldblatt and Mabb, and the root of *Anemone chinensis* Bunge. The accepted names of these plants are in accordance with The Plant List [28]. Ten milligrams of each plant were powdered using a rotary ball mill (Retch PM 100 planetary ball mill, QUB) according to manufacturer’s instructions resulting in a yield of 9 ± 0.8 mg of each plant. Two milligrams of powder were dissolved with deionised water (1:1), placed in an ultrasonic bath for 15 min then boiled in a water bath for 20 min. Solutions were stored at <4 °C prior to analysis.

### 2.3. Determination of Antibacterial Activity In Vitro

Plant extracts were screened using the broth microdilution method [29] to determine the minimum inhibitory concentration (MIC) against *L. monocytogenes, S. enteritidis,* and *E. coli.* Bacterial cultures were incubated overnight under the following conditions, optimizing them for bacterial growth: *L. monocytogenes* was incubated in tryptone soy broth (Oxoid, UK) with 5% lysed horse blood (TCS Biosciences Ltd., UK) at 35 ± 1 °C, 5% CO_2_; *E. coli,* and *S. enteritidis* were incubated in Mueller Hinton broth (Oxoid, UK) at 35 ± 1°C, ambient air. Two-fold serial dilutions were made up of antibiotics and each plant extract. One hundred microlitres of each antibiotic and plant extract concentration (2000 mg/L to 0.06 mg/L) were each added to individual wells of a 96-well plate. The bacterial culture was adjusted to an optical density equivalent to 1 × 10^8^ CFUmL^−1^ then diluted to 1 × 10^6^ CFUmL^−1^. Optical density values were confirmed with a bacterial count and were all within optical density ±0.02. One hundred microlitres of bacterial suspension was added to each well. A negative control included broth only. Ampicillin was used as a quality control for *L. monocytogenes, S. enteritidis,* and *E. coli*. The dilutions were set up in triplicate. The MIC was determined by the well with the lowest concentration of antibacterial agent that had no visible growth after incubation for 24 h under the following conditions: *L. monocytogenes* was incubated in tryptone soy agar with 5% lysed horse blood at 35 ± 1 °C, 5% CO_2_; *E. coli* and *S. enteritidis* were incubated in Mueller Hinton agar at 35 ± 1 °C, ambient air.

### 2.4. Determination of Bactericidal Activity in an In Vitro Cecum Model

Cecum contents were obtained from 3-week male Ross 308 broilers (n = 45) offered a commercial cereal-based diet (12.9 MJ/kg apparent metabolisable energy and 200 g/kg crude protein) at AFBI and stored at −80 °C. The trial was approved by the Animal Welfare Ethical Review Body at AFBI and conducted under the confines of the Animals Scientific Act 1986. One milliliter of cecum sample was mixed with 1mL selective broth to eliminate bacteria that was not the genus being studied. This was a modification to a previous method used by Johny et al. [30]. This was incubated overnight under the following conditions: *L. monocytogenes* was incubated in PALCAM broth (Sigma, UK) at 35 ± 1 °C, 5% CO_2_; *E. coli* was incubated in MacConkey broth (Sigma, UK) at 35 ± 1 °C, ambient air; and *S. enteritidis* was incubated in Tetrathionate Brilliant Green (Sigma, UK) under at 35 ± 1 °C, ambient air. Time–kill assays were based on approved methods by the Clinical and Laboratory Standards Institute [31] and modified for use in an in vitro model using cecum content as the broth. Time–kill assays were used to quantify the inhibition and killing of *E. coli*, *L. monocytogenes,* and *S. enteritidis* with plant extracts from *A. pilosa* Ledeb and *A. chinensis* Bunge (31.25 mg/L to 4000 mg/L). These plant extracts were selected for time–kill assays because they exhibited comparatively low MIC (≤ 62.5 mg/L). To conduct the time–kill assay using the in vitro model, each plant extract was added to 1mL cecum solution to obtain 1/2, 1, 2, 3, and 4 × MIC of each plant extract. Concentrations were based on the broth microdilution results which identified the minimum concentrations of plant extracts that inhibit bacteria in broth after 24 h. Separate mixtures of *L. monocytogenes, S. enteritidis,* and *E. coli* inoculum were prepared. Each mixture contained three strains of the same pathogen: *L. monocytogenes* strains QA1018, LS12519, and CP102; *S. enteritidis* strains QA60, LE103, and QA76; and *E. coli* strains UM004, UM012, and UM011. Five different bacteria colonies were selected from each of three clinical isolates per genus. These were incubated in broth overnight under the following conditions: *L. monocytogenes* was incubated in PALCAM broth under 35 ± 1 °C; 5% CO_2_; *E. coli* was incubated in MacConkey broth at 35 ± 1 °C, ambient air; and *S. enteritidis* was incubated in Tetrathionate Brilliant Green at 35 ± 1 °C, ambient air. The three isolates were each sedimented by centrifugation and the pellet of each was suspended in phosphate buffered saline (PBS) [31]. This was diluted in selective broth then added to plant extracts in cecum solution (final inoculation 1 × 10^5^ CFU/mL) and incubated overnight under the following conditions: *L. monocytogenes* was incubated in PALCAM broth at 35 ± 1 °C, 5% CO_2_; *E. coli,* was incubated in MacConkey broth at 35 ± 1 °C, ambient air; and *S. enteritidis* was incubated in Tetrathionate Brilliant Green at 35 ± 1 °C, ambient air. Replicates (n = 5) of bacterial dilutions were plated at 0, 0.5, 1, 2, 4, 6, and 24 h. Total viable count of bacteria was recorded. Distilled water was used as a negative control. Ampicillin was the quality control for *E. coli*, *L. monocytogenes,* and *S. enteritidis* [32].

### 2.5. Statistical Analysis

Minimum inhibitory concentration (MIC) and minimum bactericidal concentration (MBC) values were expressed as the mean of triplicate measurements rounded to the nearest well. MIC values were observed independently by two researchers who agreed on the value. For the percentage reduction results statistical significance in differences was measured using ANOVA completed using Prism 5.0 (Prism software available at QUB). For the time–kill results a bactericidal effect was defined as a 3-log reduction of the total viable count [31].

## 3. Results

### 3.1. Antibacterial Activity In Vitro

The MIC values of extracts of *A. pilosa* Ledeb, *S. glabra* Roxb, *A. chinensis* Bunge, and *I. domestica* (L.) Goldblatt and Mabb obtained from antibacterial testing using the broth microdilution method are presented (Table 2). The most potent antibacterial activity was exhibited by the extract of *A. pilosa* Ledeb against all *E. coli* isolates at 7.81 mg/L. The extract of *A. pilosa* Ledeb exhibited the lowest MIC and was most effective in the inhibition of *L. monocytogenes.* The extract of *A. chinensis* Bunge inhibited *S. enteritidis* at the lowest MIC of 62.5 mg/L. The extracts of *A. pilosa* Ledeb, *S. glabra* Roxb, *A. chinensis* Bunge, and *I. domestica* (L.) Goldblatt and Mabb all demonstrated considerably low MIC values ≤1000 mg/L and were therefore chosen for further investigation to determine their effect on the inhibition and killing of *L. monocytogenes*, *E. coli,* and *S. enteritidis* in an in vitro digest model.

### 3.2. Bactericidal Activity in an In Vitro Cecum Model

Next, we have investigated the bactericidal effect using an in vitro cecum model. In Table 3, Table 4 and Table 5 we present the percentage kill of bacteria incubated in cecum content over 24 h using 4 x MIC of *A. pilosa* Ledeb, *A. chinensis* Bunge, *S. glabra* Roxb, *I. domestica* Goldblatt and Mabb, and Ampicillin. All plant extracts reduced ≥ 99.9% of viable *E. coli* in ≤ 6 h (Table 3). *A. pilosa* Ledeb and Ampicillin reduced (*p* < 0.001) *E. coli* cells in 0.5 h (Table 1). *I. domestica* Goldblatt and Mabb exhibited bactericidal activity at 6 h and demonstrated a significantly (*p* < 0.001) lower percentage kill of *E. coli* than the other plant extracts at two hours. All plant extracts reduced ≥99.9% of viable *L. monocytogenes* in 24 h (Table 4). All plant extracts reduced (*p* < 0.001) *L. monocytogenes* cells in 0.5 h (Table 4). *I. domestica* Goldblatt and Mabb exhibited bactericidal activity at 24 h and demonstrated a significantly (*p* < 0.001) lower percentage kill of *L. monocytogenes* than the other plant extracts from 0.5 to 6 h. *A. chinensis* Bunge and *I. domestica* Goldblatt and Mabb reduced ≥ 99.9% of viable *S. enteritidis* in 6 h (Table 5). All plant extracts reduced (*p* < 0.001) *S. enteritidis* cells in two hours (Table 5). *I. domestica* Goldblatt and Mabb and *A. chinensis* Bunge reduced *S. enteritidis* by ≥99.9%. *S. glabra* Roxb and *A. pilosa* Ledeb exhibited the lowest reduction (*p* < 0.001) in total viable count of *S. enteritidis* (99.61% and 99.52% reduction). *A. pilosa* Ledeb was the most effective against *E. coli* and *L. monocytogenes,* while *A. chinensis* Bunge was the most effective against *S. enteritidis*.

Figure 1, Figure 2 and Figure 3 are time–kill graphs which present the total viable count of surviving populations of bacteria in the presence of extracts of *A. pilosa* Ledeb and *A. chinensis* Bunge incubated in cecum content for 24 h. At concentrations of 4 × MIC the extract of *A. pilosa* Ledeb exhibited bactericidal activity by 4 h against *L. monocytogenes* and *E. coli* (Figure 1 and Figure 2). The extract of *A. chinensis* Bunge exhibited bactericidal activity against *S. enteritidis* by 6 h (Figure 3). The MIC of these plant extracts all exhibited bacteriostatic activity and inhibited the population of pathogens over 24 h (Figure 1, Figure 2 and Figure 3).

## 4. Discussion

The aim of this research was to determine the antibacterial properties of plant extracts and evaluate the bactericidal activity of the most effective plant extracts using an in vitro poultry digest model. The broth microdilution method [29] was used to screen for antibacterial activity of *A. pilosa* Ledeb, *S. glabra* Roxb, *A. chinensis* Bunge, and *I. domestica* (L.) Goldblatt and Mabb against *L. monocytogenes, E. coli,* and *S. enteritidis*. The selection of plants was based on longstanding traditional claims [12] and current knowledge [20,33,34,35] of their antibacterial activities. A review of the literature highlighted that there is a lack of research investigating the antibacterial properties of the aqueous extracts of *A. pilosa* Ledeb, *S. glabra* Roxb, *A. chinensis* Bunge, and *I. domestica* (L.) Goldblatt and Mabb. However, the four plants chosen for this study are used traditionally as formulations or individual extracts to treat diseases of pathogenic origin such as bacterial infections and this indicates that they may exhibit antibacterial activity [12].

*Agrimonia pilosa* Ledeb exhibited the most effective antibacterial activity of all four plant extracts against both *E. coli* and *L. monocytogenes* with a comparatively low and potent MIC of 7.81 mg/L and 31.25 to 250 mg/L, respectively. *A. chinensis* Bunge exhibited the most effective antibacterial activity of all four plant extracts against *S. enteritidis,* with a comparatively-low MIC of 62.5 mg/L. Furthermore, *A. pilosa* Ledeb, *S. glabra* Roxb, *A. chinensis* Bunge, and *I. domestica* (L.) Goldblatt and Mabb demonstrated considerable antibacterial activity against Gram-positive bacteria (*L. monocytogenes*) and Gram-negative bacteria (*E. coli S. enteritidis)* with MIC values ≤ 500 mg/L. In a review of the literature Ríos and Recio [36] summarize that MIC values of ≤1000 mg/L exhibited by plant extracts are considered to demonstrate significant antibacterial activity. Concentrations above this might indicate that the bioactive compounds responsible for the antibacterial activity need to be isolated further to be effective. These results demonstrate effective antibacterial potency or a high concentration of bioactive components in these plant extracts. This is the first paper to demonstrate the potent broad-spectrum antibacterial activity of the aqueous extracts of *A. pilosa* Ledeb, *S. glabra* Roxb, *A. chinensis* Bunge, and *I. domestica* (L.) Goldblatt and Mabb against *L. monocytogenes*, *E. coli,* and *S. enteritidis.*

The broth microdilution results support the antibacterial properties of *A. pilosa* Ledeb, *S. glabra* Roxb, *A. chinensis* Bunge, and *I. domestica* (L.) Goldblatt and Mabb and provide preliminary scientific validation for the use of these plant extracts in traditional Chinese medicine to treat diseases of pathogenic origin, such as infections. This remarkable broad-spectrum antibiotic activity of *A. pilosa* Ledeb, *S. glabra* Roxb, *A. chinensis* Bunge, and *I. domestica* (L.) Goldblatt and Mabb provides compelling scientific evidence that these extracts possess antibacterial activity which may be representative of multiple bioactive compounds that inhibit bacteria. Often when single compounds are isolated from plant extracts the same bioactivity can no longer be detected [37]. Therefore, the synergistic interactions of several bioactive compounds in these plant extracts are likely to be responsible for this antibacterial activity. For example, *A. pilosa* Ledeb contains catechin and phenol derivatives which exhibit antibacterial activity against *Staphylococcus aureus* [34,38]. Catechins and phenols have been found to exert antibacterial properties against various pathogens including *E. coli* and *L. monocytogenes* [39,40,41]. These bioactive compounds could have been responsible for the antibacterial activity exhibited by the extract of *A. pilosa* Ledeb against *E. coli* and *L. monocytogenes* in this study. For example, *A. chinensis* Bunge contains quercetin and saponins [42] which exhibit antibacterial activity against various pathogens including *S. enteritidis* [43,44]. These bioactive compounds could have been responsible for the antibacterial activity exhibited by the extract of *A. chinensis* Bunge against *S. enteritidis* in this study.

The most potent antibacterial activity demonstrated by the broth microdilution results was observed using aqueous extracts of *A. pilosa* Ledeb against *E. coli* and *L. monocytogenes,* and *A. chinensis* Bunge against *S. enteritidis* using the broth microdilution method. These were selected to investigate the killing properties over time and concentration of these plant extracts whilst incubated in poultry cecum content. Research by Johny et al. [30] used autoclaved cecum content to study the effects of natural plant extracts on *Salmonella* and *Campylobacter* as the cecum content medium resembled conditions in live broilers more closely than other synthetic laboratory mediums. For this reason, poultry cecum content was used in this study. In the research by Johny et al. [30] chicken cecum contents were autoclaved to eliminate inhibitory effects of endogenous bacteria. However, the current study used the following significant modification to the model—selective broth was used to maintain the populations of existing endogenous bacteria in the ceca of poultry belonging to the genus being studied. This is more reflective of the natural environment in the poultry gastrointestinal tract because it maintains some of the background bacteria in the poultry cecum content—proteins are not denatured because autoclaving was avoided. These bacteria and proteins would be intact in live chickens and maintaining the environment of the natural content of ceca of poultry provides a model which can be used to provide results which are predictive of what might happen in the chicken gastrointestinal tract. The results of this study can therefore be used to justify further experiments and hypothesize that these plant extracts may exert an antibacterial effect against pathogenic microorganisms in vivo.

The results from the in vitro cecum model strongly support and confirm the antibacterial activity found in the broth microdilution experiment of *A. pilosa* Ledeb against *L. monocytogenes* and *E. coli*, and *A. chinensis* Bunge against *S. enteritidis*. In particular, 4 × MIC (250 mg/L and 62.5 mg/L, respectively) *A. pilosa* Ledeb significantly reduced the total viable count of *L. monocytogenes* and *E. coli* by ≥99.99% within 4 h (*p* < 0.001). *A. chinensis* Bunge significantly reduced the total viable count of *S. enteritidis* by ≥99.99% within 6 h (*p* < 0.001). This indicated rapid bactericidal activity against endogenous and inoculated bacteria cultures. At lower concentrations *A. chinensis* Bunge and *A. pilosa* Ledeb inhibited and reduced *L. monocytogenes*, *E. coli*, and *S. enteritidis.* This is the first study to report the rapid bactericidal activity of *A. pilosa* Ledeb and *A. chinensis* Bunge. This significant finding demonstrates the potent bactericidal efficacy of these plant extracts in poultry cecum content. This highlights the potential for these plant extracts to be used as alternatives to antibiotics in poultry feed to maintain pathogen populations in the poultry gastrointestinal tract. The bactericidal activity exhibited by extracts of *A. pilosa* Ledeb and *A. chinensis* Bunge indicates that caution should be taken to determine the correct concentration for supplementation in poultry feed during in vivo trials. High concentrations could potentially kill *E. coli, L. monocytogenes*, and *S. enteritidis* in poultry cecum content. Low population counts of *E. coli, L. monocytogenes*, and *S. enteritidis* have been found to be natural inhabitants of poultry cecum content without causing clinical manifestations in poultry [8,9,10]. Killing the entire *E. coli*, *L. monocytogenes*, and *S. enteritidis* population could lead to disruption of the poultry gastrointestinal tract microbiota. Modulation of the poultry gastrointestinal microbiota can therefore lead to improved health.

## 5. Conclusions

This study demonstrates broad-spectrum antibacterial activity of four plant extracts, namely, *A. pilosa* Ledeb, *S. glabra* Roxb*, A. chinensis* Bunge, and *I. domestica* (L.) Goldblatt and Mabb against *S. enteritidis*, *L. monocytogenes*, and *E. coli*. This study also provided evidence that *A. pilosa* Ledeb and *A. chinensis* Bunge reduced cecum colonization of *E. coli, L. monocytogenes,* and *S. enteritidis* in vitro. These initial results justify further in vivo trials of these plant extracts to determine their efficacy against *S. enteritidis*, *L. monocytogenes*, and *E. coli* in broiler chickens. This would also provide an opportunity to investigate the effects of *A. pilosa* Ledeb, *S. glabra* Roxb*, A. chinensis* Bunge, and *I. domestica* (L.) Goldblatt and Mabb on poultry health, performance, and microbiota. The results of this study contribute to research into the use of plant extracts in poultry feed and provide new information on the antibacterial activity of *A. chinensis* Bunge and *A. pilosa* Ledeb. They also highlight the potential of plant extracts used in traditional Chinese medicine as possible alternatives to antibiotics for use as poultry feed additives.

## Figures and Tables

**Figure 1 microorganisms-08-00962-f001:**
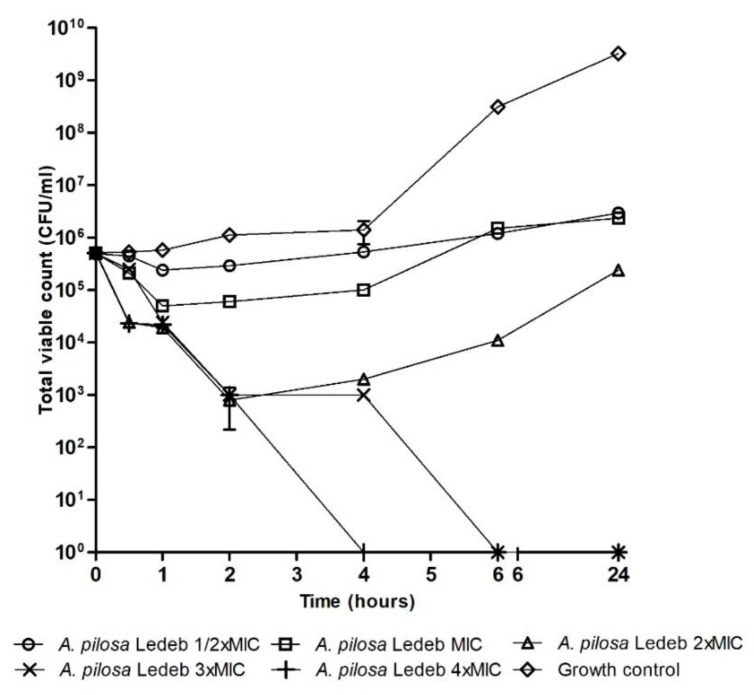
Total viable count of endogenous and inoculated *Listeria monocytogenes* QA1018, LS12519, and CP102 mixture in the presence of *Agrimonia pilosa* Ledeb.

**Figure 2 microorganisms-08-00962-f002:**
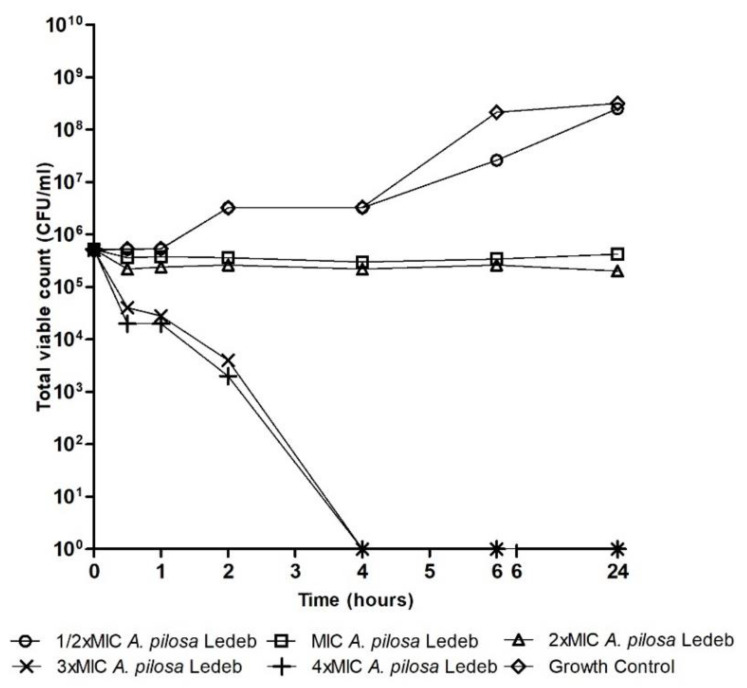
Total viable count of endogenous and inoculated *E. coli* UM004, UM011, and UM012 mixture in the presence of *Agrimonia pilosa* Ledeb.

**Figure 3 microorganisms-08-00962-f003:**
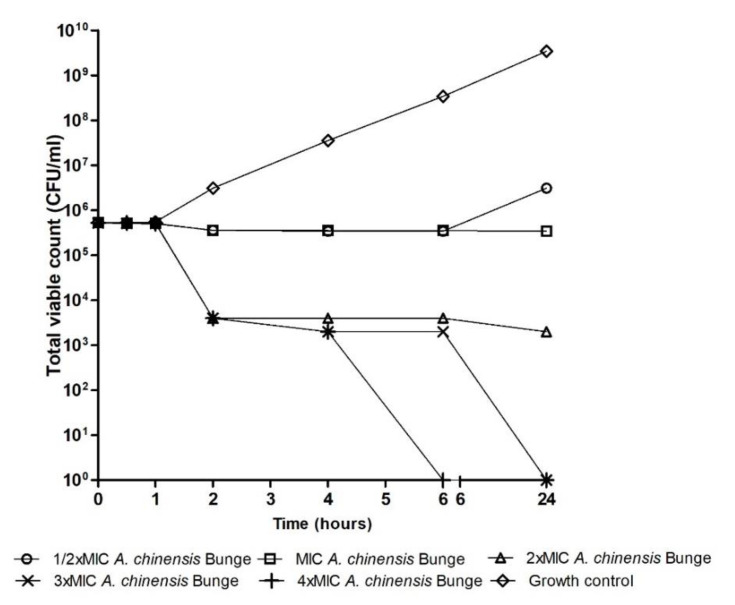
Total viable count of endogenous and inoculated *Salmonella enteritidis* QA0419, LE103, and 1F6144 mixture in the presence of *Anemone chinensis* Bunge.

**Table 1 microorganisms-08-00962-t001:** Reference strains, clinical isolates, and their sources.

Species	Isolate	Source
***Listeria monocytogenes***	NCTC 11994	Reference Queen’s University Belfast (QUB)
	LS12519	Retail ready to eat sliced meat Agri-Food and Biosciences Institute (AFBI)
	OT11230	Retail chopping board (AFBI)
	CP102	Retail ham and cheese filling (AFBI)
	CP1132	Retail cooked chicken breast (AFBI)
	QA1018	Quality assurance sample (AFBI)
***Salmonella enteritidis***	NCTC 0074	Reference (QUB)
	1F6144	Quality assurance sample (AFBI)
	LE103	Egg filter (AFBI)
	QA04/19	Quality assurance sample (AFBI)
***Escherichia coli***	ATCC 25922	Reference (QUB)
	UM004	Urinary tract infection (QUB)
	UM011	Urinary tract infection (QUB)
	UM012	Urinary tract infection (QUB)

**Table 2 microorganisms-08-00962-t002:** Minimum inhibitory concentration (MIC, mg/L) of four plant extracts relative to ampicillin against *Listeria monocytogenes, Salmonella enteritidis*, and *E. coli*.

Pathogen		Plant Extract				
		*Agrimonia pilosa* Ledeb	*Smilax glabra* Roxb	*Anemone chinensis* Bunge	*Iris domestica* (L.) Goldblatt and Mabb	Ampicillin
***L. monocytogenes***	**NCTC 11994**	31.25	62.5	125	125	0.25
	**LS12519**	31.25	31.25	125	125	0.5
	**OT11230**	62.5	125	125	125	0.5
	**CP102**	31.25	31.25	125	125	0.25
	**CP1132**	125	31.25	125	125	0.25
	**QA1018**	31.25	31.25	125	125	1
***S. enteritidis***	**NCTC 0074**	500	250	62.5	250	4
	**IF6144**	125	125	62.5	125	2
	**LE103**	125	125	62.5	125	4
	**QA0419**	125	125	62.5	125	8
***E. coli***	**ATCC 25922**	7.81	125	125	62.5	8
	**UM004**	7.81	125	125	62.5	4
	**UM011**	7.81	125	125	62.5	4
	**UM012**	7.81	125	125	62.5	4

**Table 3 microorganisms-08-00962-t003:** Average percentage kill of *E. coli* ATCC 25922, UM004, UM011, and UM012 over 24 h in the presence of 4 × minimum inhibitory concentration (MIC) of plant extracts and ampicillin over 24 h.

	Treatment						
Time (Hours)	*A. pilosa* Ledeb	*A. chinensis* Bunge	*S. glabra* Roxb	*I. domestica* (L.) Goldblatt and Mabb	Ampicillin	SEM	*p*
**0**	0	0	0	0	0	0.000	NS
**0.5**	96.24 ^b^	0.76 ^a^	0.25 ^a^	1.92 ^a^	95.10 ^b^	0.940	< 0.001
**1**	96.16 ^b^	0.76 ^a^	1.49 ^a^	2.31 ^a^	95.92 ^b^	0.890	< 0.001
**2**	99.60 ^b^	99.24 ^b^	99.27 ^b^	31.01 ^a^	99.66 ^b^	0.278	< 0.001
**4**	99.99 ^c^	99.62 ^b^	99.99 ^c^	56.59 ^a^	99.99 ^c^	0.480	< 0.001
**6**	99.99	99.99	99.99	99.99	99.99	0.000	NS
**24**	99.99	99.99	99.99	99.99	99.99	0.000	NS

^a,b,c^ superscripts indicate significance. Means with differing superscripts are significantly different (*p* < 0.001).

**Table 4 microorganisms-08-00962-t004:** Average percentage kill of *Listeria monocytogenes* NCTC 11994, QA1018, LS12519, and CP102 in the presence of 4 × minimum inhibitory concentration (MIC) of plant extracts and ampicillin over 24 h.

	Treatment						
Time (Hours)	*A. pilosa* Ledeb	*A. chinensis* Bunge	*S. glabra* Roxb	*I. domestica (L.)* Goldblatt and Mabb	Ampicillin	SEM	*p*
**0**	0	0	0	0	0	0.000	NS
**0.5**	95.47 ^c^	96.27 ^c^	93.64 ^b^	47.81 ^a^	95.97 ^c^	1.019	< 0.001
**1**	95.67 ^c^	99.63 ^d^	95.65 ^c^	43.83 ^a^	96.47 ^b^	0.358	< 0.001
**2**	99.80 ^b^	99.63 ^b^	99.61 ^b^	44.23 ^a^	99.62 ^b^	0.483	< 0.001
**4**	99.99 ^b^	99.99 ^b^	99.80 ^b^	42.63 ^a^	99.62 ^b^	0.410	< 0.001
**6**	99.99 ^b^	99.99 ^b^	99.99 ^b^	99.18 ^a^	99.99 ^b^	0.020	< 0.001
**24**	99.99	99.99	99.99	99.99	99.99	0.000	NS

^a,b,c^ superscripts indicate significance. Means with differing superscripts are significantly different (*p* < 0.001).

**Table 5 microorganisms-08-00962-t005:** Average percentage kill of *Salmonrlla enteritidis* NCTC 0074, QA0419, LE103, and 1F6144 in the presence of 4 × minimum inhibitory concentration (MIC) of plant extracts and ampicillin over 24 h.

	Treatment						
Time (Hours)	*A. pilosa* Ledeb	*A. chinensis* Bunge	*S. glabra* Roxb	*I. domestica* (L.) Goldblatt and Mabb	Ampicillin	SEM	*p*
**0**	0	0	0	0	0	0.000	NS
**0.5**	1.18 ^a^	2.26 ^a^	1.15 ^a^	0.77 ^a^	95.75 ^b^	2.559	< 0.001
**1**	1.58 ^a^	3.02 ^a^	2.32 ^a^	0.77 ^a^	96.03 ^b^	2.174	< 0.001
**2**	99.13 ^b^	99.25 ^b^	99.22 ^b^	33.59 ^a^	99.60 ^b^	0.761	< 0.001
**4**	99.29 ^b^	99.62 ^b^	99.22 ^b^	61.25 ^a^	99.80 ^b^	2.692	< 0.001
**6**	99.60 ^b^	99.99 ^c^	99.22 ^a^	99.99 ^c^	99.99 ^c^	0.132	< 0.001
**24**	99.52 ^b^	99.99^c^	99.61^a^	99.99 ^c^	99.99 ^c^	0.206	< 0.001

^a,b,c^ superscripts indicate significance. Means with differing superscripts are significantly different (*p* < 0.001).

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
