# Peer review of "Antibacterial Activity of Four Plant Extracts Extracted from Traditional Chinese Medicinal Plants against Listeria monocytogenes, Escherichia coli, and Salmonella enterica subsp. enterica serovar Enteritidis"

_microorganisms, 2020, doi:10.3390/microorganisms8060962_

Round 1

Reviewer 1 Report

English Grammar could be improved.

Some questions to consider:

Information is missing about the subspecies and serotypes of reference  Salmonella enterica. It can be important. Were all serotypes (serovars) of Salmonella enterica subs. enterica used?

Consider the use antibacterial vs bactericidal?

Line 27 “Plant extracts” or Plant Extracts?

Line 28 “Antimicrobial susceptibility” or Antimicrobial Susceptibility?

Line 32, Line 36 and so on. Revise quoting. “xxxx 13” or xxxx13?

Line 35 Change “subtheraputic” to subtherapeutic.

Lines 36-37 “Despite the legitimate high demand for antibiotics for use in the poultry sector, there are major concerns regarding their overuse and misuse.” Should not English be improved?

Line 47 Consult  with physician or veterinary if  the use of “ailment” is the best choice.

Line 56. Pharmacopeia or Pharmacopoeia?

Line 65. I. domestica (L.) Goldblatt and Mabb is native to Himalaya to Japan… Should not English be improved?

Line 93 “manufacture instructions” or manufacturer instructions?

Line 93, Line 102 and so on. Revise using ±:XX ± XX or XX±XX?

Line 100 and Line 110. 5% lysed horse blood (TCS Biosciences Ltd, UK). Is it necessary to repeat the manufacturer of 5% lysed horse blood? It is obligatory to the use the reagent of the same manufacturer for all steps.

Line 101 and Line 111. Mueller Hinton agar (Oxoid, UK). Is it necessary to repeat the manufacturer of Mueller Hinton agar? It is obligatory to the use the reagent of the same manufacturer for all steps.

Line 103, Line 105 “One hundred microlitre” or One hundred microlitres?

Line 117 “milliliter” or millitre. American English vs British English.

Lines 120, 136, 141. Is it necessary to repeat the manufacturer of PALCAM? It is obligatory to the use the reagent of the same manufacturer for all steps.

Lines 121, 136, 142. Is it necessary to repeat the manufacturer of MacConkey broth? It is obligatory to the use the reagent of the same manufacturer for all steps.

Lines 122, 137, 143. Is it necessary to repeat the manufacturer of Tetrathionate Brilliant Green? It is obligatory to the use the reagent of the same manufacturer for all steps.

Line 139 Quote (Johny et al., 2010). Use number for quote.

Line 166. “3.2. Antibacterial activity in an in vitro cecum model”. Is not it bactericidal activity in an in vitro cecum model? See Lines 172 and 175.

Line 171-172 I. domestica Goldblatt and Mabb took the longest to achieve bactericidal activity with significantly (P<0.001) lower percentage kill at two hours. Should not English be improved?

Lines 199-208 Figures 1, 2, 3. 1000 vs 100, 1001 vs 101 and so on. In my opinion 1000, 1001, 1002, 1003, 1004, 1005, 1006, 1007, 1008, 1009, 1010 should be corrected.

Line 284. Change E .coli to E. coli.

Lines 193-195 “At concentrations of 4xMIC extracts of A. pilosa Ledeb and A. chinensis Bunge exhibited bactericidal activity and there was a >3 log reduction of the total viable count of each tested pathogen”. >1 log reduction of the total viable count of each tested pathogen means >90% kill of bacteria, >2 log reduction of the total viable count of each tested pathogen means >99% kill of bacteria, >3 log reduction of the total viable count of each tested pathogen means >99.9% kill of bacteria, >4 log reduction of the total viable count of each tested pathogen means >99.99% kill of bacteria. Tables 3- 5 show A. chinensis Bunge successfully killed 99.99%  of bacteria (4 log reduction) after 24 hours. Table 3 and 4 show A. pilosa Ledeb successfully killed 99.99%  of bacteria (4 log reduction) after 24 hours also. Table 5 A. pilosa Ledeb killed 99.52% of salmonella and it means >2 log reduction, but not 3 log reduction. Is 99.52 correct number?

Line 219  MIC of 7.82mg/L? Check the number.

Line 271 62.56mg? Check the number.

Lines 297. This study’s findings contributes to research around plant extracts as alternative poultry feed additives and provides new information on A. pilosa Ledeb and A. chinensis and their antibacterial activity. Should not English be improved?

Reviewer 2 Report

This paper describes 4 plant extracts against 3 common foodborne pathogens. Overall the manuscript is well written and the methods used are appropriate. This is good preliminary data which leads onto other in vitro and in vivo studies and the data would be of interest to the readers of the journal. 

Specific comments:

L33 Is this still true? There have been massive improvements in antibiotic use in poultry farming and it is now much more controlled, I think this sentence could be re-worded to be less directly and to reflect more current practices.

L70-73 This is a repetition 

Have these extracts been used on other pathogens? Would be good to mention these in the introduction if they have.

L105 Was the OD checked with a bacterial count? How close to the OD value were they?

Table 3, 4 and 5 Is this an average of all strains or a specific strain? This needs to be made clear in the table heading

Figure 1 Can you name the species in the figure legend. 

The discussion is lacking evidence from other published studies, or is there none with these extracts, if this is the case it needs to be made clear.

The overall conclusions are good and are supported by the data presented and opens up lots of further work. 
